# Understanding Reproductive Health among Survivors of Paediatric and Young adults (URHSPY) cancers in Uganda: A mixed method study protocol

Anthony Kayiira [1,2,3] *, Daniel Zaake[2,3], Serena Xiong[4], Joyce K. Balagadde[5], Rahel Ghebre[6], Henry Wabinga[7]

1 Global Health Uganda, Kampala, Uganda, 2 Lifesure Fertility and Gynecology Centre, Kampala, Uganda, 3 Department of Obstetrics and Gynaecology, Uganda Martyrs University School of Medicine, Kampala, Uganda, 4 University of Minnesota School of Public Health, Minneapolis, Minnesota, United States of America, 5 Department of Paediatric Oncology, Uganda Cancer Institute, Kampala, Uganda, 6 Department of Obstetrics, Gynaecology and Women's Health, University of Minnesota, Minneapolis, Minnesota, United States of America, 7 Kampala Cancer Registry, Makerere University College of Health Sciences, Kampala, Uganda

* antoedwards13@gmail.com

**Data Availability Statement:** The survey questionnaires, abstraction forms and interview

## Abstract

### Background

Although reproductive failure after cancer treatment in children and young adults has been extensively described in high-income countries, there is a paucity of data in low-income settings. In addition, patient, parent, or health worker experiences, perspectives, and attitudes toward the risk of reproductive failure among young cancer patients in these settings are unknown. This study will describe the extent of reproductive morbidity associated with cancer treatment among childhood and young adult cancer survivors in Uganda. In addition, we aim to explore the contextual enablers and barriers to addressing cancer treatment-related reproductive morbidity in Uganda.

### Methods

This is an explanatory sequential mixed-method study. The quantitative phase will be a survey among childhood and young adult cancer survivors recruited from the Kampala Cancer Registry (KCR). The survey will utilize a Computer Assisted Telephone Interview (CATI) platform on a minimum of 362 survivors. The survey will obtain information on self-reported reproductive morbidity and access to oncofertility care. The qualitative phase will use grounded theory to explore contextual barriers and enablers to addressing reproductive morbidity associated with cancer treatment. The quantitative and qualitative phases will be integrated at the intermediate and results stage.

### Conclusion

Results from this study will inform the development of policy, guidelines, and programs supporting reproductive health among childhood and young adult cancer survivors.

guides are included in the supporting information files.

**Funding:** This study is funded by the North Pacific Global Health (NPGH) research fellow training consortium award to A.K, under the Fogarty International Centre and the National Institutes of Health [Grant # D43 TW009345] (https://www.fic.nih.gov/) and supported by the National Institutes of Health's National Centre for Advancing Translational Sciences [Grant # UL1 TR002494] (https://www.nih.gov/about-nih/what-we-do/nih-almanac/national-center-advancing-translational-sciences-ncats#:~:text=NCATS'%20mission%20is%20to%20catalyze,improving%20health%20through%20smart%20science.). The funders had no role in study design, data collection and analysis, decision to publish, or preparation of the manuscript.

**Competing interests:** The authors have declared that no competing interests exist.

# Background

In Uganda, the Kampala Cancer Registry recorded a 25% increase in the age-adjusted incidence of cancer over 25 years (1991–2015) [1]. The incidence of cancer in individuals aged 0–19 years is 139 per million, and overall, 3-year survival ranges from 86% for Hodgkin's disease to 30% for Wilms tumor [2]. We define young adults as individuals transitioning into adulthood and aged 18–25 years [3]. Unfortunately, the incidence of cancer and subsequent survival among young adults (18–25 years) in Uganda is largely undocumented. Nonetheless, these young cancer survivors face reproductive morbidities associated with specific chemotherapy and radiation regimens. As a result, International Society of Fertility Preservation (ISFP), American Society of Clinical Oncology (ASCO), and European Society of Medical Oncology (ESMO) recommended that fertility preservation be discussed before starting cancer treatment [4–6]. We define reproductive morbidity among cancer survivors, based on the modification of a prior definition by WHO (WHO, 1990), as any condition or dysfunction of the reproductive tract because of cancer or its treatment. These reproductive morbidities include but are not limited to amenorrhea, premature menopause, subfertility in women, and infertility in men [7–9].

The options available for fertility preservation among patients with cancer depend on age, the urgency of chemotherapy, the presence of a partner, and the availability of fertility preservation options. They vary from gonadal tissue freezing for prepubertal and teenage survivors to sperm freezing, egg freezing, and embryo freezing for young adults [4,10]. Testicular tissue freezing is still experimental and restricted to facilities with expertise. Egg and embryo freezing requires entry into an assisted reproduction cycle, which has significant cost implications. Although provided by private facilities, gamete and embryo freezing is available in Uganda. However, prepubertal and teenage survivors have no available option because of the lack of expertise in tissue cryopreservation in Uganda. Nonetheless, due to limited awareness and knowledge gap on cancer-related reproductive morbidity, there is little or no fertility preservation done for cancer patients in Uganda.

There is a lack of local data on the prevalence of cancer treatment-related reproductive morbidity among childhood and young adult cancer survivors. In addition, for cancer treatment-related reproductive morbidity to be addressed, childhood cancer survivorship care needs to be grounded in context-specific experiences and expectations of cancer patients, healthcare workers, and guardians. Understanding how these key stakeholders experience the cancer treatment journey and perceive engagement of reproductive concerns could inform the implementation of client-centred oncofertility programs. It is equally critical to explore reproductive expectations, decision-making processes, and supports (if any) that exist during the interaction between children with cancer, guardians, and health workers.

## Aims and objectives

This mixed-method study aims to describe reproductive health outcomes among survivors of childhood and young adult cancer and follow up with a comprehensive contextualized explanation of the experiences, expectations, and perceptions of parents or guardians and health workers regarding cancer treatment-related reproductive morbidity as they engage through the cancer care continuum.

Quantitative objectives: (1) describe the burden of self-reported reproductive morbidity; (2) describe access to reproductive counseling and fertility preservation among survivors of childhood and young adult cancer.

Qualitative objectives: (1) describe participants' reproductive expectations during cancer care; (2) describe participants' knowledge of cancer treatment-related reproductive morbidity;

(3) qualitatively map participant's engagement on reproductive concerns throughout the cancer care continuum; (4) identify and describe critical interactions in the decision-making process; (5) describe the perception and attitude towards cancer treatment-related reproductive morbidity; (6) identify contextual and personal influences that shape reproductive expectations during the cancer care continuum; (7) conceptually model the interaction between guardians and health workers, regarding cancer treatment-related reproductive morbidity, including core determinant processes, existing support structures, and coping pathways.

## Method

### Study design

An explanatory sequential mixed-method design. In explanatory sequential mixed method design, a quantitative method is conducted to give the general scope of the problem. The subsequent qualitative approach is built on or modified based on the quantitative data and adds insight and context to the statistical results [11–13]. In this study, the quantitative phase is a population-based survey of childhood and young adult cancer survivors. We shall follow up with the qualitative phase, based on grounded theory (GT) methodology, conducting in-depth interviews among purposefully selected guardians of children with cancer and health workers involved in cancer care. Priority will be given to the qualitative phase of the data [11] since we seek to gain a contextualized understanding of the interactive healthcare process that underlies the reported reproductive health outcomes. At the synthesis stage, the resulting substantive theory of the process will help to explain the survey results.

The Uganda Cancer Institute (UCI) and the University of Minnesota (UMN) institutional review boards reviewed and approved the study. The survey participants will provide verbal consent, which will be documented and archived in the REDCap informed consent framework. The in-depth interview participants will provide written informed consent.

### Quantitative phase

A population-based survey of survivors of childhood and young adult cancer. This will capture population-level data about self-reported reproductive morbidity and access to fertility preservation. The population will consist of childhood and young adult cancer survivors, diagnosed in 10 years i.e., between 2007 and 2018. A Computer Assisted Telephone Interview (CATI) will be utilized. CATI is a robust, reliable method of collecting community-level health-related data [13–15].

The Kampala Cancer Registry (KCR) is the region's oldest, curated, and complete source of cancer incidence data. The KCR collects data on cancer incidence for the population of Kyadondo county, which includes the city of Kampala—the capital of Uganda—and a peri-urban area extending 30 Km to the North. Kyadondo county lies on the equator at a longitude of approximately 340 E and covers an area of 1914 km2. The population is estimated at 2,614,994.

**Survey population.**   The survey population will be survivors of pubertal, adolescent, and young adult cancer diagnosed between 2007 and 2018. Participants eligible for the survey must comply with all the following at selection from the registry: current age $\geq$ 18 years, age at cancer diagnosis (0–25 years), alive at the time of last contact, and with complete registry information, including a traceable phone contact. We included individuals 0–25 to encompass childhood and young adult cancer survivors, using a young adult definition of 18–25 years. Although there is significant variation between societies for appropriate age definition of young adulthood, the Society for Adolescent Health and Medicine defines young adulthood as a period ranging from 18–25 years of age [14]. We assume, among others [3,14,15], that this

transition represents youth [16] and is associated with several unique physiological, social, and psychological phenomena that render them vulnerable to receiving inadequate health care. In addition, for many within this age range, their reproductive expectations are not concrete, and their interest in reproductive survivorship is high.

To estimate the sample size for the cross-sectional survey, we used the results of the Childhood Cancer Survivor Study (CCSS) cohort [17]. In this study, the proportion of survivors that had been pregnant or fathered a pregnancy was 38% compared to 62% of sibling controls. Therefore, using the ability to have or father a pregnancy as a crude indicator of reproductive function, we applied 38% as a prevalence of normal reproductive function among childhood cancer survivors in the statistical formula described by J Charan and T Biswas [18] and determined the minimum sample size for the survey to be 362 childhood and young adult cancer survivors.

**Survey instrument.** An extensive literature review of the MEDLINE database was performed to find relevant structured questionnaires that could answer the study objectives. Priority was given to questionnaires administered by an interviewer and used the CATI method. The Furthering Understanding of Cancer, Health, and Survivorship in Adult (FUCHSIA) Women's Study [19] questionnaire was identified as the most suitable to answer the research question. The Understanding Reproductive Health among Survivors of pediatric and young adult (URHSPY) CANCER questionnaire was developed by the study team from the FUCHSIA questionnaire. This was achieved by modification, deletion, and addition of questions that are context-sensitive and addressing the current study objectives. In addition, a male version was designed and made suitable to understand reproductive health among male cancer survivors. The study team reviewed the draft URHSPY questionnaire to ensure conciseness, relevance, and completeness. The URHSPY cancers CATI will address these areas: (1) cancer treatment history, (2) demographics, (3) menstrual health, (4) infertility history, (5) pregnancy history, and (6) lifestyle. The final version of the URHSPY questionnaire was programmed into REDCap data capture software [20].

Further pre-testing of complex skip patterns, typing errors, non-offensiveness, question flow, and completeness were done by an expert on the study team. Two research assistants were trained on how to conduct telephone-assisted interviews. A field pre-test of the CATI-based survey was done on a sample of 10 cancer survivors accessed conveniently with a similar eligibility profile as the intended survey subjects. After the pre-test, the questionnaire was modified before its final rollout based on any feedback concerning the appropriateness, question interpretation, question intrusiveness, and interview time.

**Survey data collection, management, and analysis.** All eligible subjects were contacted to participate in the survey. In addition, a text message containing summarised information about the ongoing survey were sent to the registered KCR contacts of all potential subjects before the first call by the study team.Participants were contacted by telephone using the phone contact provided in the registry. If contact was successful, the interviewer introduced themselves, the survey, and its relevance. The interviewer ascertained the identity of the contact and confirmed their vital status. All efforts have been made to agree on the most convenient time and day of the week for the interview. The interviewer contacted the participants on the agreed time and day, which may or may not be the same day as the first contact. Upon making contact, the interviewer re-introduced themselves and provide more information about the survey. The interviewer then administered and guided the participant through the computer-assisted verbal consent process and the survey. The interviewer read the questions directly from the interactive computer screen on the survey instrument and recorded the verbal answers of the respondent. The research assistants were trained to observe emotional cues during the telephone survey.

In addition, the participants have the option to decline to answer questions that are sensitive or halt the interview. In case of emotional distress, the participant is referred to the study counselor for further care. To define a failed contact attempt, the investigators agreed that ten call attempts made over five working days with at least two calls per day (morning and afternoon) were exhaustive considering available resources and would be the desired criterion All participants get a digital or USSD airtime gift card worth UGX 20,000, to compensate for their time at the end of the interview. In addition, the principal investigator does a data check of 10% of the interviews performed off-site via online REDCap project administrator access. This is a data quality check, after which a weekly data quality report is made and uploaded onto a secure, password-protected research purpose-only computer. When reporting this protocol, 50% of the intended participants have been contacted.

All REDCap survey data will be exported into up-to-date STATA 17, StataCorp. 2021. Stata Statistical Software: Release 17. College Station, TX: StataCorp LLC. Descriptive statistics will be performed on the data to estimate various proportions and means of survey responses. Further analysis to draw inferences from the responses in each survey section will be done using the chi-squared test for categorical responses and the T-test for continuous responses. We will use regression methods with appropriate interaction terms for subgroup analyses between men and women. We will calculate Relative Risk (RR) and RR Reductions (RRR) with corresponding 95% confidence intervals to compare dichotomous responses. The difference in means will be used for additional analysis of continuous responses. For all tests, we will use 2-sided p-values with alpha $\leq 0.05$ level of significance.

## Qualitative phase

We will utilize GT methodology. GT is based on the symbolic interactionist theory, which states that people construct realities from the meaning they place on interactions with their surroundings [21]. This process is continuous, and individuals adapt their reality depending on their changing interactions [21,22]. Therefore, GT is a comprehensive qualitative method that systematically collects and analyses data to develop a central theory that explains and or predicts a given phenomenon [21,23]. This gives the researcher an understanding and control of the phenomenon, which can be operationalized in hypothesis testing [23]. In this study, we aim to explain the process of interactions between guardians and health workers regarding cancer treatment-related reproductive morbidity during the cancer care continuum.

GT is a recursive method that originates from the data, which is comparatively analyzed to generate conceptual categories that derive the underlying theory [21,23]. This is done through an iterative process of theoretical sampling, where theoretically relevant context-rich data sources are purposefully selected, and data is then collected and analyzed to generate meaningful categories [21,23]. These in turn inform guiding questions and subsequent sampling in an "inductive-deductive cycle" [24]. This cycle is repeated while constantly comparing and refining emerging theoretical constructs into more abstract versions [21]. This is continued until no nuance, meaning, or theory emerges from each new data sampling cycle to refine either conceptual category [23,24]. This is termed "theoretical saturation" and is determined by the data's empirical limits, integration, the density of the derived conceptual categories, and the researcher's theoretical sensitivity [23].

This study will employ the synthesized approach to GT described by H-Y Chen and JR Boore [24]. This multi-step synthesized approach is based on Strauss and Corbin's style of theoretical sampling and memoranda writing, Strauss and Corbin's and Glaser's definition of theoretical sensitivity, Charmaz's memo-writing and data collection, Glaser's wide range of selected theoretical coding families, and Glaser and Strauss' criteria for theoretical saturation [24].

The study team consists of an early career researcher on a global health research fellowship (author 1), three clinical researchers with expertise in reproductive medicine, gynecology oncology, and pediatric oncology (authors 2,4 and 5), a Ph.D. candidate with expertise in qualitative methodology including GT (author 3), and an expert in cancer epidemiological surveillance (author 6).

We chose Uganda Cancer Institute (UCI) since it is Uganda's only tertiary cancer treatment facility. Available options for cancer treatment at UCI include chemotherapy, radiotherapy, and surgery. In addition, it is the only center in Uganda providing dedicated pediatric oncology services, with an estimated 600 new pediatric cancer cases per year. Unfortunately, there is currently no dedicated childhood cancer survivorship or oncofertility program.

**In-depth interview population.** Two categories of participants will be interviewed: (1) Guardians older than 18 years and attending to children (0–18 yrs) diagnosed with cancer and who are out of the induction and consolidation phases of treatment. (2) the oncology care team includes oncology physicians, medical officers, and nurses.

The sample size will be determined by theoretical saturation [23] i.e., when no new meaning can be obtained from additional interviews. Based on GT methodology, we shall use theoretical sampling on a varied purposeful sample of participants. Deviating from the tenets of GT [23], we estimated sample size priori using the assumption by MM Hennink, et al. [25] that it would take 16–24 interviews for inducted or deducted codes to reach meaning saturation. We plan to enrol 20 participants as guardians of children. Comprehensive data analysis will inform the composition of the subsequent sample. If theoretical saturation is not reached, additional interviews will be performed until at least three further interviews do not generate new meaningful codes. We shall interview all ten health workers attached to the pediatric oncology department. This sample includes oncology fellows, oncologists, oncology nurses, and medical officers.

**Data collection.** An in-depth interview is the best way to add depth and context to the population-level data. Therefore, separate guides have been developed for guardians and health workers. The interviews will be conducted with an interview guide (S1 and S2 Files) with inherent flexibility. The guide is systematic and semi-structured, with open-ended questions, and not limiting the participant's expression of their experience. The guide will be developed with significant input from the quantitative survey results [11]. In addition, the interviewers will follow probes during the dialogue with participants to encourage the articulation of experiences, perceptions, and attitudes. The interview guides were pre-tested on a group of five eligible participants to address question clarity, acceptability, and understanding.

After enrolment, face-to-face interviews will be held in a comfortable and private setting. All participants must consent to the study before the interview can start. At all times, an effort will be made to maintain comfort, and alleviate stress and privacy during the interview process, as described by R Elmir, et al. [26]. The interviewers are research nurses certified in good clinical practice and basic human research subject protection, with training in conducting in-depth interviews for qualitative research. The interview will be recorded with a Sony® IC recorder (model: ICD-PX470). Field notes will be taken during the interview to document observed nonverbal behaviours. All participants will get compensation worth UGX 40,000 to compensate for their time in the study. For cases where there is recognizable emotional fallout from the interview, the interview will be halted and postponed. The participant will then be connected to a study counselor to follow up with the debrief.

**Data management and analysis.** All the recordings will be transcribed verbatim and translated into English in Microsoft word® version 16.41. Each transcription will be made within 24 hours of ending the interview. A professional research transcription service will do this. The resulting transcripts will be analyzed in qualitative data analysis software NVivo

version 12 by QRS international. Using theoretical sensitivity and inductively coding for whichever meaning arises from the data [23], each transcript will be coded thrice by a team of three investigators, using the participant's own words, as GT requires [23]. A synthesized method of GT will be utilized in a multi-step systematic manner, as described by [24]. In sequential order starting with line-by-line in vivo coding using the participant's own words, shorter word phrases will then be developed that capture the meaning of the participant's words while grouping similar coded phrases; concepts are named; concepts are grouped and refined into categories; theoretical coding families will be used to isolate concepts and connect them in different ways; new connections of categories will be established; subcategories will be defined; linkage among categories will be developed; the core category defined; the core category leading to the discovery of the basic social process identified; and lastly explanatory frameworks leading to the derivation of a substantive theory developed.

To reach an inter-coder consensus, we shall use methods described by MA Cascio, et al. [27]. In addition, to ensure rigor in the study process, we will have reflexivity among authors, critical feedback among authors, peer examination, peer briefing, using participant's own words, consistency in the interview guide across interviews, and an audit trail including data collection process, coding, consensus guidelines, and memos throughout the research process [28].

## Discussion

Studies in high-income settings show that individual, system, and environmental factors affect the perspectives and attitudes of health workers toward the risk of reproductive failure among young cancer patients [29–31]. Unfortunately, the burden and associated barriers or enablers to the favorable patient, guardian, or health worker perspectives and attitudes toward the risk of reproductive failure among young cancer patients in Uganda are unknown. This will be the first study in Uganda to describe the burden and explore contextual factors that may impact reproductive morbidity associated with cancer treatment among childhood and young adult survivors.

We describe the self-reported reproductive failure and the experience of oncofertility among survivors of childhood and young adult cancer. The substantive theory of the process resulting from the qualitative phase will add clarity and explain these statistical results. This study will contribute salient aspects to understanding the context of reproductive morbidity in children and young adults with cancer. This will stir up the much-needed conversation about fertility preservation in children and young adults with cancer. Furthermore, insights from this study will be used in recommendations for survivorship care and in developing an oncofertility program grounded in this social-cultural context.

Inherently, mixed-method studies are performed to answer complex questions beyond the scope of one design method [11]. This study provides a situational analysis of the burden of reproductive morbidity among childhood and young adult cancer survivors while offering insights into the institutional and patient factors that may explain the limited effort to address this. In addition, the study applies CATI based survey, which leverages telecommunication to obtain data that would otherwise be logistically impossible, more so during the current COVID-19 pandemic. The use of CATI-based surveys in the middle to low-income settings has been validated and found comparable to other traditional interview modalities [32,33]. Nonetheless, the CATI-based survey will be limited by recall bias and misclassification of self-reported outcomes, which can introduce ascertainment bias. The absence of a counterfactual group of individuals without cancer also limits the inferences' validity. Lastly, study results grounded in data, through systematically exploring experiences, knowledge, perceptions, and

attitudes toward cancer treatment-related reproductive morbidity, will help elucidate the underlying theory. From this, we hope to build on further inquiry and inform implementation programs directed toward addressing cancer treatment-related reproductive morbidity among children with cancer.

## Conclusion

Results from this study will provide a foundation for awareness and advocacy for fertility preservation in at-risk young cancer patients in Uganda. In addition, the study will provide valuable insight and justification for incorporating reproductive survivorship into national cancer control plans.

## Supporting information

**S1 File.**
(DOCX)

**S2 File.**
(DOCX)

**S3 File.**
(ZIP)

## Acknowledgments

We acknowledge the generous contribution of Penelope P Howards the PI of the FUCHSIA group, for granting us access to the FUCHSIA CATI research instrument. This was ultimately the backbone for designing the URHSPY cancers CATI.

## Author Contributions

**Conceptualization:** Anthony Kayiira, Daniel Zaake, Serena Xiong, Joyce K. Balagadde, Henry Wabinga.

**Data curation:** Serena Xiong, Rahel Ghebre.

**Formal analysis:** Serena Xiong, Rahel Ghebre.

**Funding acquisition:** Anthony Kayiira, Rahel Ghebre, Henry Wabinga.

**Investigation:** Anthony Kayiira, Serena Xiong.

**Methodology:** Anthony Kayiira, Daniel Zaake, Serena Xiong, Joyce K. Balagadde.

**Project administration:** Anthony Kayiira, Daniel Zaake, Joyce K. Balagadde, Rahel Ghebre.

**Resources:** Daniel Zaake, Joyce K. Balagadde, Rahel Ghebre.

**Software:** Serena Xiong.

**Supervision:** Daniel Zaake, Serena Xiong, Joyce K. Balagadde, Rahel Ghebre, Henry Wabinga.

**Validation:** Daniel Zaake, Joyce K. Balagadde, Rahel Ghebre, Henry Wabinga.

**Writing – original draft:** Anthony Kayiira.

**Writing – review & editing:** Anthony Kayiira, Daniel Zaake, Serena Xiong, Joyce K. Balagadde, Rahel Ghebre, Henry Wabinga.

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
