## [Decision Letter · Decision Letter 0]

26 Sep 2022

PONE-D-22-20391Understanding Reproductive Health among Survivors of Paediatric and Young adults (URHSPY) Cancers in a low-income setting: a mixed method study protocolPLOS ONE

Dear Dr. Kayiira,

Thank you for submitting your manuscript to PLOS ONE. After careful consideration, we feel that it has merit but does not fully meet PLOS ONE’s publication criteria as it currently stands. Therefore, we invite you to submit a revised version of the manuscript that addresses the points raised during the review process.

We look forward to receiving your revised manuscript.

Kind regards,

Callam Davidson

Editorial Office

PLOS ONE

 "This study is funded by the North Pacific Global Health (NPGH) research fellow training consortium award to A.K, under the Fogarty International Centre and the National Institutes of Health [Grant # D43 TW009345] (https://www.fic.nih.gov/) and supported by the National Institutes of Health's National Centre for Advancing Translational Sciences [Grant # UL1 TR002494] (https://www.nih.gov/about-nih/what-we-do/nih-almanac/national-center-advancing-translational-sciences-ncats#:~:text=NCATS'%20mission%20is%20to%20catalyze,improving%20health%20through%20smart%20science.). The funding organisations did not have any role in the design of the study or writing the manuscript. They will not be involved in data collection, analysis, and interpretation of data."

Reviewers' comments:

Reviewer's Responses to Questions

**Comments to the Author**

1. Does the manuscript provide a valid rationale for the proposed study, with clearly identified and justified research questions?

Reviewer #1: Yes

Reviewer #2: Yes

2. Is the protocol technically sound and planned in a manner that will lead to a meaningful outcome and allow testing the stated hypotheses?

Reviewer #1: Yes

Reviewer #2: Yes

3. Is the methodology feasible and described in sufficient detail to allow the work to be replicable?

Reviewer #1: Yes

Reviewer #2: Yes

4. Have the authors described where all data underlying the findings will be made available when the study is complete?

Reviewer #1: Yes

Reviewer #2: No

5. Is the manuscript presented in an intelligible fashion and written in standard English?

Reviewer #1: No

Reviewer #2: Yes

6. Review Comments to the Author

You may also provide optional suggestions and comments to authors that they might find helpful in planning their study.

Reviewer #1: The authors present a study protocol of mixed methods study. The rationale for the study is clearly described. The results of this study will provide novel information about the reproductive health climate of young cancer patients living in Uganda.

Overall, the protocol manuscript would significantly benefit from editorial assistance for grammar and punctuation corrections in each paragraph. It is also repetitive in some sections. Here are some specific suggestions for revisions:

Abstract Methods: Please include the estimated sample size for both the survey and interview components.

Background: This section would benefit from a re-organization to concisely present the rationale for the study’s purpose. The first paragraph can be removed as it is not necessary to present the global cancer incidence. Suggestion to start the background section with the incidence of cancers in Uganda among young cancer survivors and the reproductive morbidity (consider merging current paragraph 2 and the note definition of reproductive morbidity).

Purpose statement: The specific objectives are repetitive with what is already stated in the paragraph – consider merging all this information into one paragraph.

Methods Survey Approach:

Setting: This paragraph is very long; suggestion to shorten the amount of detailed information on the Kyadondo County and its residents.

Eligibility Criteria: Suggestion to state the inclusion and exclusion criteria in sentence format versus a bulleted list. Please also include your rationale for only including cancer patients diagnosed from ages 0 to 25 years. The National Cancer Institute definition of adolescent and young adult cancer patient includes ages 15 to 39 years. Please explain why you are not including cancer patients aged 26 to 39 years.

Sampling: This section is not clear; please consider revising for clarity.

Recruitment strategy: Please explain how the message with the survey will be sent to potential subjects prior to the first call. Will information be mailed?

Survey data collection, management and analysis: The paragraphs describing the survey instruments and data collection are quite lengthy and somewhat repetitive. Consider revising to be more concise. For the data collection processes, please provide a rationale for making 10 attempts and up to 2 per day for the first contact.

Methods Qualitative Approach:

Setting: This paragraph is too long; suggestion to shorten the amount of detailed information in this paragraph.

Eligibility Criteria: Suggestion to state the inclusion and exclusion criteria in sentence format versus a bulleted list.

In-depth interview instruments: Please move the sentences describing the interview guides located in the “Data collection process” paragraph into this section to provide more details regarding the semi-structured interview questions.

Data collection process: This section can be shortened for better flow and clarity.

Qualitative data management and analysis: Please provide more details on how many individuals will be coding the qualitative transcripts and how consensus between coders will be reached.

Trustworthiness: Consider removing this section from the manuscript.

Discussion: Consider revising the last three paragraphs into one study strengths and limitations paragraph.

Reviewer #2: Thank you for inviting me to review such an interesting and thought-provoking research proposal. With the information provided by the authors, I believe there is enough justification for the study. Although the methodology appears appropriate, I still have some concerns. My first concern borders on the sampling method for the cross-sectional arm of the study - Will the authors recruit only the calculated sample size of 362 from the identified study population, or do they plan to recruit all consenting eligible survivors? If the former is the case, then the authors should calculate and adjust for non-response, define as well how they will select the sample population from the study population. My second concern relates to the questionnaire - was it adapted to the local population. Thirdly, is there any compensation for the participants for their time? Fourthly, the authors noted the possibility of adverse psychological effects by the study - How do they plan to monitor this? or do they have a plan for interim assessment of the magnitude of such psychological harm? Finally, the authors should be specific on how the study will influence policy and practice. RECOMMENDATION: Approved after addressing the concerns raised in the review.

7. PLOS authors have the option to publish the peer review history of their article (what does this mean?). If published, this will include your full peer review and any attached files.

Reviewer #1: No

Reviewer #2: **Yes: **Ekwuazi Kingsley Emeka

---

## [Author Response · Author response to Decision Letter 0]

11 Nov 2022

Wednesday, November 2, 2022

Editor-in-Chief

PLOS ONE journal

Dear sir or madam,

RE: Response to reviewers for protocol # PONE-D-22-20391; Understanding Reproductive Health among Survivors of Paediatric and Young adults (URHSPY) Cancers in Uganda: a mixed method study protocol

Thank you for accepting to review our manuscript. We have prepared a point-by-point response to the reviewers concerns in the body on this letter. Changes in the manuscript are tracked, and where needed highlighted in red. 

Journal requirements

1. Comment: When submitting your revision, we need you to address these additional requirements. Please ensure that your manuscript meets PLOS ONE's style requirements, including those for file naming. The PLOS ONE style templates can be found at 

Response: The manuscript style and file naming has been modified to meet PLOS ONE's requirements. 

2. Comment: Thank you for stating the following financial disclosure: "This study is funded by the North Pacific Global Health (NPGH) research fellow training consortium award to A.K, under the Fogarty International Centre and the National Institutes of Health [Grant # D43 TW009345] (https://www.fic.nih.gov/) and supported by the National Institutes of Health's National Centre for Advancing Translational Sciences [Grant # UL1 TR002494] (https://www.nih.gov/about-nih/what-we-do/nih-almanac/national-center-advancing-translational-sciences. ncats#:~:text=NCATS'%20mission%20is%20to%20catalyze,improving%20health%20through%20smart%20science.). The funding organizations did not have any role in the design of the study or writing the manuscript. They will not be involved in data collection, analysis, and interpretation of data." Please state what role the funders took in the study. If the funders had no role, please state: "The funders had no role in study design, data collection and analysis, decision to publish, or preparation of the manuscript." If this statement is not correct you must amend it as needed. 

Response: We have amended the financial disclosure statement to include: “The funders had no role in study design, data collection and analysis, decision to publish, or preparation of the manuscript.” [see page 14, under funding section].

3. Comment: Your ethics statement should only appear in the Methods section of your manuscript. If your ethics statement is written in any section besides the Methods, please move it to the Methods section and delete it from any other section. Please ensure that your ethics statement is included in your manuscript, as the ethics statement entered into the online submission form will not be published alongside your manuscript.

Response: The ethics section has been moved it to the Methods section and deleted from any other section. [see page 6, first section/paragraph].

REVIEWER 1

1. Comment: Is the manuscript presented in an intelligible fashion and written in standard English? PLOS ONE does not copyedit accepted manuscripts, so the language in submitted articles must be clear, correct, and unambiguous. Any typographical or grammatical errors should be corrected at revision, so please note any specific errors here. Reviewer #1: No.

Response: Extensive efforts have been made to edit grammar and spelling throughout the manuscript. The paragraphs have also been reordered and revised to succinctly convey the message of the manuscript. 

2. Comment: Overall, the protocol manuscript would significantly benefit from editorial assistance for grammar and punctuation corrections in each paragraph. It is also repetitive in some sections. 

Response: Extensive efforts have been made to edit grammar and spelling throughout the manuscript. The paragraphs have also been reordered and revised to succinctly convey the message of the manuscript.

3. Comment: Abstract Methods: Please include the estimated sample size for both the survey and interview components.

Response: The sample size for both study designs have been included in the manuscript abstract. [see page 2, under methods, lines 5 and 8] 

4. Comment: Background: This section would benefit from a re-organization to concisely present the rationale for the study’s purpose. The first paragraph can be removed as it is not necessary to present the global cancer incidence. Suggestion to start the background section with the incidence of cancers in Uganda among young cancer survivors and the reproductive morbidity (consider merging current paragraph 2 and the note definition of reproductive morbidity).

Response: The background section has been revised and rearranged to convey a concise message. [see page background section, pages 3-5]

5. Comment: Purpose statement: The specific objectives are repetitive with what is already stated in the paragraph – consider merging all this information into one paragraph.

Response: The purpose statement and specific objectives have been revised and merged into one paragraph in the background. [see background, paragraph 5]

6. Comment: Methods Survey Approach:

Setting: This paragraph is very long; suggestion to shorten the amount of detailed information on the Kyadondo County and its residents. 

Response: Setting subsection in survey method has been revised. [see setting, under survey approach]

7. Comment: Eligibility Criteria: Suggestion to state the inclusion and exclusion criteria in sentence format versus a bulleted list. Please also include your rationale for only including cancer patients diagnosed from ages 0 to 25 years. The National Cancer Institute definition of adolescent and young adult cancer patient includes ages 15 to 39 years. Please explain why you are not including cancer patients aged 26 to 39 years.

Response: Formatted and revised eligibility criteria. Included rationale for choosing 0-25 years. Although, there is significant variation between societies for appropriate age defining young adulthood, the Society for Adolescent Health and Medicine defines young adulthood as a period ranging 18-25 years of age [16]. We assume among others [4, 16, 17], that this transition represents youth [18] and is associated with several unique physiological, social, and psychological phenomena that render them vulnerable to receiving inadequate health care. In addition, for many within in this age range, their reproductive expectations are not concrete as such they are a target group for reproductive survivorship compared to individuals 26-39 years of age. [see page 6, Eligibility section lines 5-16]

8. Comment: Sampling: This section is not clear; please consider revising for clarity.

Response: Revised to concisely describe how we arrived at the sample size for the survey. [see page 7, sample size section]

9. Comment: Recruitment strategy: Please explain how the message with the survey will be sent to potential subjects prior to the first call. Will information be mailed?

Response: A text message containing summarised information about the ongoing survey will be sent to the registered contacts of all potential subjects before the first call by the study team. [see page 8, under recruitment strategy]

10. Comment: Survey data collection, management, and analysis: The paragraphs describing the survey instruments and data collection are quite lengthy and somewhat repetitive. Consider revising to be more concise. For the data collection processes, please provide a rationale for making 10 attempts and up to 2 per day for the first contact.

Response: Survey data collection, management, and analysis revised and rearranged to a more concise version. To define a failed contact attempt, the investigators agreed that 10 call attempts made over 5 working days with at least two calls per day (morning and afternoon), was exhaustive considering available resources and would be the desired criterion. [see page 8-9, Survey data collection, management, and analysis section]

11. Comment: Methods Qualitative Approach:

Setting: This paragraph is too long; suggestion to shorten the amount of detailed information in this paragraph.

Response: Setting for qualitative approach revised and formatted. [see page 10, under setting section]

12. Comment: Eligibility Criteria: Suggestion to state the inclusion and exclusion criteria in sentence format versus a bulleted list.

Response: Formatted eligibility section. [see page 10, under eligibility section]

13. Comment: In-depth interview instruments: Please move the sentences describing the interview guides located in the “Data collection process” paragraph into this section to provide more details regarding the semi-structured interview questions. Data collection process: This section can be shortened for better flow and clarity. 

Response: The sections In-depth interview instruments and Data collection process have been revised and rearranged to convey a concise message. [see page 10-11]

14. Comment: Qualitative data management and analysis: Please provide more details on how many individuals will be coding the qualitative transcripts and how consensus between coders will be reached.

Response: Each transcript will be coded thrice, by a team of three investigators. To reach inter-coder consensus we shall use methods described by MA Cascio, et al. [27]. All axial codes and their representative open codes for all coders will be tabled and compared to identify discrepancies at the level of the axial codes. The axial codes are then sorted into overlapping and exact match categories, any residual points of non-consensus are discussed until unanimous resolution is reached. [ see page 11-12, under section Qualitative data collection, management, and analysis]

15. Comment: Trustworthiness: Consider removing this section from the manuscript.

Response: This section was deleted. 

16. Comment: Discussion: Consider revising the last three paragraphs into one study strengths and limitations paragraph.

Response: The discussion has been revised and rearranged. [see page 13, under discussion]

REVIEWER 2

1. Comment: Have the authors described where all data underlying the findings will be made available when the study is complete? The PLOS Data policy requires authors to make all data underlying the findings described in their manuscript fully available without restriction, with rare exception, at the time of publication. The data should be provided as part of the manuscript or its supporting information or deposited to a public repository. For example, in addition to summary statistics, the data points behind means, medians and variance measures should be available. If there are restrictions on publicly sharing data—e.g., participant privacy or use of data from a third party—those must be specified. Reviewer #2: No

Response: The survey questionnaires, abstraction forms and interview guides are included in the supporting information files. [see page 14, Availability of data and materials]

2. Comment: My first concern borders on the sampling method for the cross-sectional arm of the study - Will the authors recruit only the calculated sample size of 362 from the identified study population, or do they plan to recruit all consenting eligible survivors? If the former is the case, then the authors should calculate and adjust for non-response, define as well how they will select the sample population from the study population

Response: We plan to recruit all consenting eligible survivors. 362 is the only the minimum sample required to answer the objective. 

3. Comment: My second concern relates to the questionnaire - was it adapted to the local population

Response: The survey questionnaire was adapted to the local consent and pre-tested before rolling out.

4. Comment: Thirdly, is there any compensation for the participants for their time?

Response: There is a compensation of 20,000 UGX for the survey and 40,000 UGX for the in-depth interview. [see page 8, paragraph 1, line 8 and page 12, paragraph, line 6]

5. Comment: Fourthly, the authors noted the possibility of adverse psychological effects by the study - How do they plan to monitor this? or do they have a plan for interim assessment of the magnitude of such psychological harm?

Response: The research assistants will be trained to observe emotional cues during the telephone survey and face to face in-depth interviews. In addition, the participants will have an option decline answering questions that are sensitive or halt the interview if they have overwhelming emotional fall out. In case of emotional fall out the participant will be referred to the study counsellor for further care. There will be interim assessment of the magnitude of such psychological harm. [see page 9, paragraph 1, line 3-6]

6. Comment: Finally, the authors should be specific on how the study will influence policy and practice.

Response: Results from this study will provide a foundation for awareness and advocacy for fertility preservation in at risk young cancer patients in Uganda. In addition, the study will provide valuable insight and justification for incorporating reproductive survivorship into the national cancer control plans. [see page 14, conclusion section]

Thank you so much. Look forward to the feedback.

Kind regards

Anthony Kayiira 

Corresponding author

---

## [Decision Letter · Decision Letter 1]

31 Jan 2023

PONE-D-22-20391R1Understanding Reproductive Health among Survivors of Paediatric and Young adults (URHSPY) Cancers in Uganda: a mixed method study protocolPLOS ONE

Dear Dr. Kayiira,

Thank you for submitting your manuscript to PLOS ONE. After careful consideration, we feel that it has merit but does not fully meet PLOS ONE’s publication criteria as it currently stands. Therefore, we invite you to submit a revised version of the manuscript that addresses the points raised during the review process.

We look forward to receiving your revised manuscript.

Kind regards,

Dorina Onoya

Academic Editor

PLOS ONE

Journal Requirements:

Reviewers' comments:

Reviewer's Responses to Questions

**Comments to the Author**

1. Does the manuscript provide a valid rationale for the proposed study, with clearly identified and justified research questions?

Reviewer #1: Yes

Reviewer #3: Partly

2. Is the protocol technically sound and planned in a manner that will lead to a meaningful outcome and allow testing the stated hypotheses?

Reviewer #1: Yes

Reviewer #3: Partly

3. Is the methodology feasible and described in sufficient detail to allow the work to be replicable?

Reviewer #1: Yes

Reviewer #3: No

4. Have the authors described where all data underlying the findings will be made available when the study is complete?

Reviewer #1: Yes

Reviewer #3: Yes

5. Is the manuscript presented in an intelligible fashion and written in standard English?

Reviewer #1: Yes

Reviewer #3: Yes

6. Review Comments to the Author

You may also provide optional suggestions and comments to authors that they might find helpful in planning their study.

Reviewer #1: The authors response to reviewers comments was thorough and addressed the primary concerns to strengthen their study protocol manuscript. The results of this study will provide novel information about the reproductive health climate of pediatric and young adult cancer patients living in Uganda.

The revised manuscript is a significant improvement from the previous version. Since PLOS ONE does not copyedit accepted manuscripts, a suggestion is to conduct a thorough copy edit of the manuscript to correct for double words, spelling, grammar and punctuation throughout the manuscript. For example, in the Background section, first paragraph, fifth sentence, “Incidence” should not be capitalized and in the 6th sentence, “Nonetheless” is listed twice. Another example is in the Discussion section, second paragraph, second sentence, “CATI based surgery” should be “CATI based survey”.

Reviewer #3: General: The research resulting from studies done on this protocol will contribute with very valuable information regarding reproductive health. The protocol is fairly concise, I was not reviewer of the first version but have gone through all actions from previous reviewers. The responses appear adequate.

By having experience and expertise in qualitative methodology as additional reviewer my focus has caught some additional points that would benefit from being addressed. My biggest concern is that by reading your protocol you have neither convinced me that the study has a mixed methods design, nor that Grounded Theory is adequately planned to be used. It seems more like two different studies are planned, if not you need to better reflect why this is a mixed methods study and have data being generated will be analysed together and provide added value to each other. Grounded Theory is a complex method to use, and I cannot see that the objective or the description of the methodology is in line with how the method should be used. Some choices needs to be made regarding these issues, and then it must be described with consistency.

1. Keywords in abstract should be ordered alphabetically

2. Page 6 “Eligibility criteria”, I would suggest you to reconsider the age defining “young adults”, 18-39 would be more appropriate. See reference from The National cancer institute in the USA, They define young adults between 15-39, and from legal motives it may be simpler to include those from 18. “The national Cancer Institute. Adolescents and Young Adults with Cancer. Available online: https://www.cancer.gov/types/aya

3. Page 6 heading “Sample size”, incomplete sentence; In this study, the proportion of survivors that had or sired a pregnancy…”

4. Not convinced that what you are doing is Grounded Theory, a form of mixed methods: Yes, but the qualitative plan could more easily be understood if you would keep it o have a qualitative approach where interviews are planned to be analysed using a thematic method. If you choose to keep GT as the preferred method, it must be clear if you are supposed to develop theory, or study the processes in relation to the subject – which presently is NOT a part of the described objective and you are not presenting any adequate reference to the method of GT.

5. Page 10; “In depth interview instruments”: Preferably I would like to see you interview guide, the sample questions suggest that those being interviewed are supposed to be those that have suffered from cancer – but from the objective, they are supposed to be guardians – unclear, please revise. (For some reason i cannot access supplements)

6. Doing qualitative interviews without being a researcher trained in the methods is not suitable – especially when using GT, the interviewer must have training in GT in order to make interviews that follows the process of GT. And also the same if it is supposed to be narrative interviews analysed with thematic analysis

7. Similarly regarding the “mixed methods design”, here in the protocol it is ok, but in the resulting article you must argue how the data analysis and presentation will be done in line with how it is expected to be done in mixed design. No adequate reference for mixed methods designs is present.

a. My suggestion is that when you are panning to publish, that you separate it into two articles, since the mixed design is weak and the objectives of the two parts differentiate too much.

7. PLOS authors have the option to publish the peer review history of their article (what does this mean?). If published, this will include your full peer review and any attached files.

Reviewer #1: No

Reviewer #3: **Yes: **Mats Jong

---

## [Author Response · Author response to Decision Letter 1]

17 Mar 2023

Friday, March 17, 2023

Editor-in-Chief

PLOS ONE journal

RE: Response to reviewers for protocol # PONE-D-22-20391; Understanding Reproductive Health among Survivors of Pediatric and Young adults (URHSPY) Cancers in Uganda: a mixed method study protocol

Thank you for accepting to review our manuscript. We have prepared a point-by-point response to the reviewers concerns in the body on this letter. Changes in the manuscript are tracked, and where needed highlighted in red. 

Journal requirements

Comment: Please review your reference list to ensure that it is complete and correct. If you have cited papers that have been retracted, please include the rationale for doing so in the manuscript text or remove these references and replace them with relevant current references. Any changes to the reference list should be mentioned in the rebuttal letter that accompanies your revised manuscript. If you need to cite a retracted article, indicate the article’s retracted status in the References list and also include a citation and full reference for the retraction notice.

Response: The reference list has been reviewed and found to have no retracted papers. Any changes to the number references, are in line with changes to the manuscript text. i.e., removing or adding reference citations. 

REVIEWER 1

Comment: The authors response to reviewers’ comments was thorough and addressed the primary concerns to strengthen their study protocol manuscript. The results of this study will provide novel information about the reproductive health climate of pediatric and young adult cancer patients living in Uganda. The revised manuscript is a significant improvement from the previous version. Since PLOS ONE does not copyedit accepted manuscripts, a suggestion is to conduct a thorough copy edit of the manuscript to correct for double words, spelling, grammar, and punctuation throughout the manuscript. For example, in the Background section, first paragraph, fifth sentence, “Incidence” should not be capitalized and in the 6th sentence, “Nonetheless” is listed twice. Another example is in the Discussion section, second paragraph, second sentence, “CATI based surgery” should be “CATI based survey”.

Response: Thank you for your comment. Extensive efforts have been made to edit grammar, punctuation and spelling throughout the manuscript. The paragraphs have also been reordered and revised to succinctly convey the message of the manuscript. 

REVIEWER 3

Comment: By having experience and expertise in qualitative methodology as additional reviewer my focus has caught some additional points that would benefit from being addressed. My biggest concern is that by reading your protocol you have neither convinced me that the study has a mixed methods design, nor that Grounded Theory is adequately planned to be used. It seems more like two different studies are planned, if not you need to better reflect why this is a mixed methods study and have data being generated will be analyzed together and provide added value to each other. Grounded Theory is a complex method to use, and I cannot see that the objective or the description of the methodology is in line with how the method should be used. Some choices needs to be made regarding these issues, and then it must be described with consistency.

Response: Thank you for your comment. We have made necessary edits and additions to the overall protocol to reflect the explanatory sequential mixed method design and an elaborative description of the grounded theory methodology. Please see the sections; aims and objective under background, study design under method, and qualitative phase.

Comment: Keywords in abstract should be ordered alphabetically.

Response: Keywords ordered alphabetically.

Comment: Page 6 “Eligibility criteria”, I would suggest you to reconsider the age defining “young adults”, 18-39 would be more appropriate. See reference from The National cancer institute in the USA, They define young adults between 15-39, and from legal motives it may be simpler to include those from 18. “The national Cancer Institute. Adolescents and Young Adults with Cancer. Available online: https://www.cancer.gov/types/aya

Response: Thank you for your comment. We do acknowledge the age category for young adults outlined by the NCI is 15-39yrs. We chose 0-25 to encompass childhood and young adult cancer survivors, using a young adult definition of 18-25 years. We also acknowledge there is significant variation between societies for appropriate age definition of young adulthood, the Society for Adolescent Health and Medicine defines young adulthood as a period ranging 18-25 years of age [14]. We assume among others [3, 14, 15], that this transition represents youth [16] and is associated with several unique physiological, social, and psychological phenomena that render them vulnerable to receiving inadequate health care. These psychosocial factors are less likely to manifest among individuals >25 but <40 years.

Our main assumption is that young adults older than 25 years at diagnosis in our population are likely to have already had a child (this is based on national demographic health survey data). Considering our study is aimed at children and young adults who fall into the category of reproductive vulnerability i.e. likely not to have a family and likely to have unrealized reproductive expectations, we excluded individuals diagnosed with cancer older than 25 years. 

Comment: Page 6 heading “Sample size”, incomplete sentence; In this study, the proportion of survivors that had or sired a pregnancy…”

Response: Necessary edits made. See sample size sub-section, line 3.

Comment: Not convinced that what you are doing is Grounded Theory, a form of mixed methods: Yes, but the qualitative plan could more easily be understood if you would keep it o have a qualitative approach where interviews are planned to be analyzed using a thematic method. If you choose to keep GT as the preferred method, it must be clear if you are supposed to develop theory, or study the processes in relation to the subject – which presently is NOT a part of the described objective, and you are not presenting any adequate reference to the method of GT.

Response: Thank you for your comment. We have made necessary edits and additions to the overall protocol to reflect the explanatory sequential mixed method design and an elaborative description of the grounded theory methodology. Please see the sections; aims and objective under background, study design under method, and qualitative phase.

Comment: Page 10; “In depth interview instruments”: Preferably I would like to see you interview guide, the sample questions suggest that those being interviewed are supposed to be those that have suffered from cancer – but from the objective, they are supposed to be guardians – unclear, please revise. (For some reason i cannot access supplements)

Response: Thank you for the comment. This an edit error. We have revised and removed the sample questions. Instead, we have included them in the supplement interview guides. (have attached these to the revision).

Comment: Doing qualitative interviews without being a researcher trained in the methods is not suitable – especially when using GT, the interviewer must have training in GT in order to make interviews that follows the process of GT. And also the same if it is supposed to be narrative interviews analyzed with thematic analysis

Response: Thank you for the comment. The first author is research fellow undertaking qualitative methodology. Significant time has spanned from the time of the initial protocol submission. During this time the author has gone through rigorous training in grounded theory and analysis. The third author has completed her PhD thesis and with considerable expertise in grounded theory methodology. 

Comment: Similarly, regarding the “mixed methods design”, here in the protocol it is ok, but in the resulting article you must argue how the data analysis and presentation will be done in line with how it is expected to be done in mixed design. No adequate reference for mixed methods designs is present. My suggestion is that when you are planning to publish, that you separate it into two articles, since the mixed design is weak and the objectives of the two parts differentiate too much.

Response: We have made efforts to revise the protocol to best suit mixed method design, including furnishing with appropriate references. Our hope was that we perform a situation analysis with a population that has had a longer exposure time for the outcome of interest. Unfortunately, the long exposure time negates recall of circumstances surrounding treatment especially in survivors who suffered cancer at ages less than 18 years. The best population for the qualitative part would be current patients going through the cancer treatment. In this case parents being the decision makers for minors. If the data collected from both methods cannot be integrated, we welcome the idea of publishing separate manuscripts. 

Thank you so much. Looking forward to the feedback.

Kind regards

Anthony Kayiira 

Corresponding author

---

## [Editor Report · Decision Letter 2]

13 Apr 2023

Understanding Reproductive Health among Survivors of Paediatric and Young adults (URHSPY) Cancers in Uganda: a mixed method study protocol

PONE-D-22-20391R2

Dear Dr. Anthony Kayiira,

We’re pleased to inform you that your manuscript has been judged scientifically suitable for publication and will be formally accepted for publication once it meets all outstanding technical requirements.

Kind regards,

Dorina Onoya

Academic Editor

PLOS ONE

Additional Editor Comments (optional):

The revised manuscript has adequately addressed the reviewer's comments and recommendations. However, it is recommended that the authors conduct a final review to address any minor typographical errors. Additionally, there are a couple of abbreviations that should be spelled out. While it may not be necessary to go through another round of reviews, the authors should make sure to address these issues during the proofreading stage before publication.
---

## [Editor Report · Acceptance letter]

16 Apr 2023

PONE-D-22-20391R2 

Understanding Reproductive Health among Survivors of Paediatric and Young adults (URHSPY) Cancers in Uganda: a mixed method study protocol 

Dear Dr. Kayiira:

I'm pleased to inform you that your manuscript has been deemed suitable for publication in PLOS ONE. Congratulations! Your manuscript is now with our production department. 

Kind regards, 

on behalf of

Dr. Dorina Onoya 

Academic Editor

PLOS ONE